

# Study on a hybrid algorithm combining enhanced ant colony optimization and double improved simulated annealing via clustering in the Traveling Salesman Problem (TSP)

Tan Hao, Wu Yingnian, Zhang Jiaxing and Zhang Jing

School of Automation, Beijing Information Science and Technology University, Beijing, China

## ABSTRACT

In the process of solving the Traveling Salesman Problem (TSP), both Ant Colony Optimization and simulated annealing exhibit different limitations depending on the dataset. This article aims to address these limitations by improving and combining these two algorithms using the clustering method. The problems tackled include Ant Colony Optimization's susceptibility to stagnation, slow convergence, excessive computations, and local optima, as well as simulated annealing's slow convergence and limited local search capability. By conducting tests on various TSPLIB datasets, the algorithm proposed in this article demonstrates improved convergence speed and solution quality compared to traditional algorithms. Furthermore, it exhibits certain advantages over other existing improved algorithms. Finally, this article applies this algorithm to logistics transportation, yielding excellent results.

## INTRODUCTION

Since the Traveling Salesman Problem (TSP) is widely used in artificial intelligence, logistics transportation, circuit board design, and other fields (*Di Placido, Archetti & Cerrone, 2022*; *Crişan, Pintea & Palade, 2017*; *Yu, Lian & Yang, 2021*; *Wang et al., 2016*; *Lim, Kanagaraj & Ponnambalam, 2014*; *Li et al., 2018*), it has been studied by a large number of scholars. Some researchers are currently using exact algorithms such as the branch-and-bound algorithm, mixed integer linear programming method, and dynamic programming method to solve the TSP instances (*Dell'Amico, Montemanni & Novellani, 2021*; *Gelareh et al., 2020*; *Lu, Benlic & Wu, 2018*). However, as instances become more complex and data sets become larger, the exact algorithms no longer have advantages. Instead, various approximate algorithms are more suitable for solving the instances that are complex and the data is huge.

The bionic algorithm is a heuristic algorithm that simulates natural phenomena or processes. Many research scholars have used bionic algorithms, such as the genetic algorithm, particle swarm optimization and so on to approximate the solution of TSP

Corresponding author
Wu Yingnian, wuyingnian@126.com

instances (*Zheng et al., 2023*; *Zheng, Zhang & Yang, 2022*; *Al-Gaphari, Al-Amry & Al-Nuzaili, 2021*; *Khan & Maiti, 2019*; *Zhong et al., 2018*). Ant Colony Optimization and simulated annealing also are considered suitable for solving the TSP.

In nature, the foraging habits of ants served as the basis principle for the Ant Colony Optimization (ACO) (*Colorni, Dorigo & Maniezzo, 1991*). The Italian researcher, M. Dorigo, based it on the fact that ants can usually choose an optimal route between their nest and food supply. In the years since the proposal of the Ant Colony Optimization, numerous scholars have shown interest in enhancing its performance and have put forth various approaches for improvement (*Dorigo, Maniezzo & Colorni, 1996*; *Stutzle & Hoos, 2000*; *Dong, Guo & Tickle, 2012*). Even today, many researchers continue to refine and propose diverse methods to enhance the ACO. This sustained interest and ongoing efforts in improving the ACO are primarily driven by its significant potential for solving complex optimization problems efficiently and effectively. The researchers discovered that ants release a chemical known as a pheromone when foraging. The ACO uses this pheromone as a cue for its pathfinding direction and always proceeds in the direction of higher pheromone concentration. The ACO is characterized by excellent robustness, rapid high accuracy, strong local exploration capabilities, and distributed parallel computing for small-scale solutions. In large-scale instances, the algorithm performs poorly because it is prone to stalling, slow convergence, excessive computing effort, and falling into local optimal solutions. Scholars have suggested a variety of improved methods to solve these flaws. Many academics believe that refreshing the pheromone update methods in the ACO is an area for development. The primary reason is that the pheromone is essential to the ACO (*Du et al., 2022*; *Ning et al., 2021*, *2018*). *Du et al. (2022)* propose a method to correct pheromone levels, which encourages ants to deposit pheromones in nearby cities, enabling them to select superior cities. *Ning et al. (2021)* propose a pheromone matrix with negative feedback. This way expands the diversity of the way that ants choose links and places links that have not been visited before inferior links, so the effectiveness of the ants at constructing paths is improved dramatically. In order to enhance the global search capability, *Ning et al. (2018)* propose a novel pheromone smoothing mechanism designed to reinitialize the pheromone matrix when the ACO's search process approaches a defined stagnation state. However, since pheromones are among the quantities produced by the model, the enhanced algorithms that depend on them cannot overcome the model's restrictions. Meanwhile, in the TSP instances, the ACO uses a roulette wheel strategy to choose the next city to visit and then travels across every city. Therefore, the pheromone strategy of the improved ACO is also unable to avoid the selection of poor path points that lead the algorithm to the local optimal solution. Some scholars have found the shortcomings of the optimal search strategy and framework of the ACO and have proposed their methods (*Gao, 2020*; *Gülcü et al., 2018*; *Wei, Han & Hong, 2014*; *Ratanavilisagul, 2017*). *Gao (2020)* propose a new ACO that utilizes a strategy of combining pairs of searching ants to diversify the solution space. Additionally, to reduce the influence of having a limited number of meeting ants, a threshold constant is introduced. This method improves the solution accuracy and reduces the work of the ant colony. *Gülcü et al. (2018)* propose using the 3-opt operator as a means to improve the

quality of the ACO's solution. *Wei, Han & Hong (2014)* the authors propose embedding the ACO into the cultural algorithm framework using a dual inheritance mechanism to make the optimal solution evolve in the population space and belief space. Although these methods improve the quality of the solution, they prolong the time to solve the problem by applying complex search strategies, which leads to stagnation of the algorithm in dealing with large-scale TSP instances. Many other scholars have found that the shortcomings of the ACO can be compensated for using a combination of other algorithms (*Gong et al., 2022*; *Wang & Han, 2021*; *Rokbani et al., 2021*; *Yang et al., 2020*; *Gulcu et al., 2018*; *Qian & Su, 2018*; *Gunduz, Kiran & Ozceylan, 2015*); they obtain excellent solutions for the TSP. *Gong et al. (2022)* propose a hybrid algorithm based on a state-adaptive slime mold model and fractional-order ant system (SSMFAS) to address the TSP. *Wang & Han (2021)* propose the hybrid symbiotic organisms search (SOS) and ACO (SOS-ACO). *Gulcu et al. (2018)* propose a parallel cooperative hybrid algorithm for solving the TSP instances.

Physical annealing served as the basis for the simulated annealing (SA) (*Metropolis et al., 1953*) concept. This concept was used in the discipline of combinatorial optimization by *Kirkpatrick, Gelatt & Vecchi (1983)*. In the years since the proposal of the SA, numerous scholars have shown interest in enhancing its performance and have put forth various approaches for improvement (*Allwright & Carpenter, 1989*; *Lin, Kao & Hsu, 1993*; *Geng et al., 2011*). The SA is a global search algorithm with flexible, widespread, efficient operation, less initial condition requirements, and other advantages. Currently, many researchers have found that the search strategy and parameter tuning of the SA can be challenging. As a result, they have proposed various improvement methods to address these difficulties (*Wang et al., 2015*; *Zhao, Xiong & Shu, 2015*; *Lin, Bian & Liu, 2016*). *Wang et al. (2015)* propose a multi-agent SA with instance-based sampling (MSA-IBS) by exploiting the learning ability of instance-based search algorithms to solve TSP instances. *Zhao, Xiong & Shu (2015)* propose a SA with a hybrid local search for the TSP, which improves solution accuracy. *Lin, Bian & Liu (2016)* propose a hybrid SA—tabu search algorithm to solve the TSP. Fully considering the characteristics of the hybrid algorithm, they develop a dynamic neighborhood structure for the hybrid algorithm to improve search efficiency by reducing the randomness of the conventional 2-opt neighborhood. More scholars have found that the SA can be combined with other algorithms to provide a better result. The popular hybrid SA for the TSP is the list-based SA. This novel hybrid algorithm mainly combines SA with the list-based threshold accepting (LBTA) algorithm. It has proven to be an effective solution for solving large-scale TSP instances (*Zhan et al., 2016*; *Wang et al., 2019*; *Ilin et al., 2022*; *Ilhan & Gokmen, 2022*). Other researchers have also discovered that combining SA with other algorithms not included in the LBTA also can yield promising results and enhance the solving capability of the SA (*Deng, Xiong & Wang, 2021*; *He, Wu & Xu, 2018*; *Ezugwu, Adewumi & Frîncu, 2017*). *Deng, Xiong & Wang (2021)* propose a hybrid Cellular Genetic Algorithm with the SA (SCGA), which is closer to the theoretical optimal value and has good robustness. *He, Wu & Xu (2018)* combine SA and the genetic algorithm to propose the Improved Genetic Simulated Annealing (IGSAA), This method makes SA more effective in avoiding getting stuck in local optima. The symbiotic biological search algorithm and the SA were merged in the literature

(*Ezugwu, Adewumi & Frîncu, 2017*) to increase the accuracy of the solution and speed up the convergence of the SA.

Firstly, in this article, we introduce two strategies to address slow convergence and susceptibility to local optima of the ACO, which effectively accelerate its convergence and enhance the accuracy to a considerable extent. Secondly, we propose two strategies to overcome the accuracy limitations of the simulated annealing, which traditionally lacked optimization ability. Subsequently, we analyze the shortcomings of the two improved algorithms and synergistically combine their advantages to devise a new hybrid algorithm. We thoroughly test this algorithm using 22 different TSP instances. Additionally, we conduct a comprehensive comparison with other traditional algorithms and state-of-the-art techniques from the literature. Finally, we apply this novel algorithm to the domain of logistics and transportation, showcasing its potential and practicality in real-world scenarios.

# BASIC INTRODUCTION AND IMPROVEMENT OF BASIC ALGORITHM

## Description of the traveling salesman problem

In the DFJ formulation, the TSP can be represented by an assignment-complete graph $G = (V, E)$, where $V$ represents the set of vertices and $E$ represents the set of edges. The distance between vertices $i$ and $j$ is denoted as $d_{i,j}$ and is assumed to be known. To represent the TSP mathematically, we introduce binary decision variables $x_{ij}$, where $x_{ij} = 1$ if the edge $(i, j)$ is included in the loop path, and $x_{ij} = 0$ otherwise. The objective is to minimize the total distance traveled, which can be expressed as:

$$\min Z = \sum_{i=1}^{n} \sum_{j=1}^{n} d_{ij} x_{ij} \tag{1}$$

Subject to the following constraints. Each city must be visited exactly once:

$$s.t. \sum_{j=1}^{n} x_{ij} = 1 \; i \in V \tag{2}$$

Each city must be left exactly once:

$$\sum_{i=1}^{n} x_{ij} = 1 \; j \in V \tag{3}$$

Subtour elimination constraints to prevent subtours:

$$\sum_{i \in S} \sum_{j \in S} x_{ij} \leq |S| - 1, \forall S \subset V, 2 \leq |S| \leq n - 1 \tag{4}$$

$$x_{ij} \in \{0, 1\}, i, j \in V \tag{5}$$

In these constraints, $S$ represents a subset of cities, and $|S|$ denotes the cardinality of set $S$. The subtour elimination constraints ensure that the solution does not contain any subtours (partial cycles).

## K-Medoids algorithm (K-M)

The K-Medoids algorithm (*Ezugwu, Adewumi & Frîncu, 2017*) is an exploratory classification algorithm that seeks to take the 'median' in each cluster as the centroid, each centroid is a sample point of the data set. Compared to the K-means algorithm, the K-M has good robustness to anomalous data in the sample data, weakening the effect of outliers on the overall clustering algorithm and reducing the bias of the clustering results, making it more accurate and suitable for data used to deal with the TSP.

## Introduction and improvement of the elite ant colony optimization

### The elite ant colony optimization

The ACO has been under development for over 20 years, and numerous researchers have been continuously improving and refining ACOs. Some notable improved ACO include the Maximum-Minimum Ant Colony Optimization, Elite Ant Colony Optimization, Sorting-Based Ant Colony Optimization, and others. Among them, the Elite Ant Colony Optimization (EACO) (*Dorigo, Maniezzo & Colorni, 1996*) introduces an elite ant strategy, which rewards ants that discover the optimal path in the current cycle with additional pheromones. This strategy reduces the number of iterations required by the ACO and improves the quality of the solution to some extent.

(1) Transition probability

The ant colony uses a probabilistic selection method to decide to transfer from the current city $i$ to the next city $j$ and releases a certain amount of pheromone during the transfer process. In the initial process, the pheromone concentration of each path is equal, then the transfer probability of an ant to transfer from the current city $i$ to the next city $j$ is shown in Eq. (6).

$$P_{ij}^k = \begin{cases} \dfrac{[\theta_{ij}]^\alpha [\eta_{ij}]^\beta}{\sum\limits_{r \notin tabu_k} [\theta_{ir}]^\alpha [\eta_{ir}]^\beta} & j \notin tabu_k \\ 0 & otherwise \end{cases} \tag{6}$$

$\eta_{ij}$ is the heuristic factor between the current city $i$ and the next city $j$, $\theta_{ij}$ is the pheromone concentration left by the ant between the current city $i$ and the next city $j$, $tabu_k [k = 1, 2, 3...]$ is called the tabu list to record the cities that ant $k$ has currently traveled, $\alpha$ is the pheromone factor, meaning the importance of path with remaining pheromones, $\beta$ is the heuristic factor, denoting the affection of heuristic information.

(2) Update of pheromones

The pheromones on each road are updated once all the ants have traveled through all the cities. The three steps of the EACO's pheromone update are pheromone volatilization, ant release of pheromones along their separate paths, and pheromone reward for elite ants.

Pheromone volatilization equation:

$$\tau_{ij} = (1 - \ell)\tau_{ij}(t) \tag{7}$$

$\ell$ is representative of the rate of volatilization.

The number of pheromones remains on the path at the current iteration for ant $k$, which can be calculated as:

$$\Delta t_{ij}^{k}(t) = \begin{cases} \frac{Q}{d_{(ij)}} & (i,j) \in T^{k} \\ 0 & \text{otherwise} \end{cases} \tag{8}$$

$Q$ is the pheromone augmentation factor, $T^{k}$ is the path of the ant $k$, $d_{(ij)}$ is the length of the current edge, when $d_{(ij)}$ smaller, more pheromones will be obtained on the current edge.

Update formula for additional pheromones awarded to elite ants:

$$\Delta t_{ij}^{best}(t) = \begin{cases} \frac{e}{L^{best}} & (i,j) \in T^{bk} \\ 0 & \text{otherwise} \end{cases} \tag{9}$$

$e$ is the pheromone augmentation factor for the optimal path, $L^{best}$ is the current loop optimal solution, $T^{bk}$ is the path of the elite ants.

The pheromones for all ants are updated using the following equation:

$$\tau_{ij}(t+1) = (1 - \ell)\tau_{ij}(t) + \sum_{k=1}^{m} \Delta t_{ij}^{k}(t) + \Delta t_{ij}^{best}(t) \tag{10}$$

### Adaptive elite ant colony optimization (AEACO)

The EACO exhibits similar robustness to the ACO and can be easily integrated with other algorithms. While the EACO improves upon the number of iterations required by the ACO, it still inherits the limitations of slow convergence and susceptibility to local optima. Therefore, this article proposes two improvement strategies for the slow convergence of the EACO.

#### Strategy one

The optimal path between a colony's exploration of a nest and a food source relies heavily on the information transmitted through pheromones. If the ant colony already knows the starting city $a$ and the ending city $b$ (the ending city is the city before the ant colony returns to its starting point), the ant colony tends to choose the city closer to the ending city, leading to a biased selection process for the next visited city during exploration. Utilizing the best pheromone information between the starting city $a$ and the ending city $b$, the ant colony can efficiently determine the appropriate direction to explore, thereby accelerating the search process. To address this, this article proposes an improved pheromone update strategy in conjunction with the EACO. Furthermore, all edges are initialized based on the distance between the starting city $a$ and the ending city $b$. The concentration of pheromones in the initialized ant colony is determined as follows:

$$\tau_{ij}(0) = \frac{d_{ab}}{d_{aj}} + d_{jb} \tag{11}$$

$\tau_{ij}(0)$ is the initial pheromone concentration between the current city $i$ and the next city $j$, $d_{ab}$ is the linear distance between the starting city $a$ and the ending city $b$, $d_{aj}$ is the distance from the starting city $a$ to the next city $j$, $d_{jb}$ is the distance from the next city $j$ to the ending city $b$.

This strategy changes the initial pheromone concentration of the EACO and focuses on the distance between the current city and the ending city, which provides directional guidance for the initial ant colony and avoids blind searches of the ant colony, thus improving the speed of solution and accuracy of the solution.

*Strategy two*

The method for ants to select the next city from the current city is primarily based on the roulette wheel betting method. This method ensures a well-balanced algorithm, where ants with higher fitness values are more likely to be selected, while ants with lower fitness values still have a chance to be chosen. This approach allows the ant colony to explore and experiment within the solution space, allowing all unvisited cities to be selected. However, in the actual solution process, it is important to avoid consecutively visiting two cities that are particularly far apart. If the ant colony selects a distant city as the next destination from its current location, it would result in wasted iterations, leading to increased solution time for the algorithm. More critically, it could significantly affect the overall direction of the algorithm's solution and potentially trap it in a local optimal solution. Therefore, this article restricts the city selection process to the roulette wheel betting method. Ants are not allowed to choose a more distant city as the next visited destination. This constraint is expressed mathematically as follows:

$$R = \frac{\lambda \sum_{r \notin tabu_k} d_{ir}}{Number\ of\ cities\ not\ visited} \tag{12}$$

$$x_{ir} = \begin{cases} 1 & d_{ir} < R \\ 0 & d_{ir} \geq R \end{cases} \tag{13}$$

$R$ is the maximum distance from other cities that can be visited, $\lambda$ is parameter [1,2], $d_{ir}$ is the current city $i$ to a city $r$ that has not been visited, Eq. (13) indicates whether the unvisited city $r$ can be added to the roulette match, 1 is acceptance and 0 indicates no acceptance, $x_{ir}$ is the decision variable for the current city $i$ to the unvisited city $r$.

Our algorithm called the Adaptive Elite Ant Colony Optimization (AEACO) is an efficient optimization-seeking algorithm for small-scale TSP instances, which will automatically adjust the number of iterations and the number of ants for different small-scale instances. Meanwhile, a comparison test with the ACO and the AEACO is run to evaluate the performance of the AEACO. The AEACO, ACO, and AEACO are put to the test 30 times, with each algorithm's optimal solution, average error rate, and solution time is provided. Table 1, Figures 1 and 2 show the experimental results, Meanwhile, *Time* is the

**Table 1 Results of the ACO, EACO and AEACO for solving small-scale instances.**

| Instances | Opt | Iterations | Algorithm | Best | $SD_{avg}$ (%) | Time (s) |
|---|---|---|---|---|---|---|
| bays29 | 2,020 | m = 29, itermax = 40 | ACO | 2,053 | 5.9 | 2.87 |
| | | m = 29, itermax = 30 | EACO | 2,087 | 3.4 | 2.35 |
| | | m = 29, itermax = 15 | AEACO | 2,020 | 0.3 | 0.85 |
| berlin52 | 7,542 | m = 52, itermax = 70 | ACO | 8,349.5 | 12.9 | 27.81 |
| | | m = 52, itermax = 60 | EACO | 7,853.4 | 4.8 | 23.68 |
| | | m = 52, itermax = 26 | AEACO | 7,612.39 | 1.7 | 8.83 |
| pr76 | 108,159 | m = 76, itermax = 150 | ACO | 126,151.2 | 18.3 | 170.91 |
| | | m = 76, itermax = 100 | EACO | 120,576.7 | 13.5 | 126.67 |
| | | m = 76, itermax = 38 | AEACO | 119,813.4 | 12.2 | 39.44 |

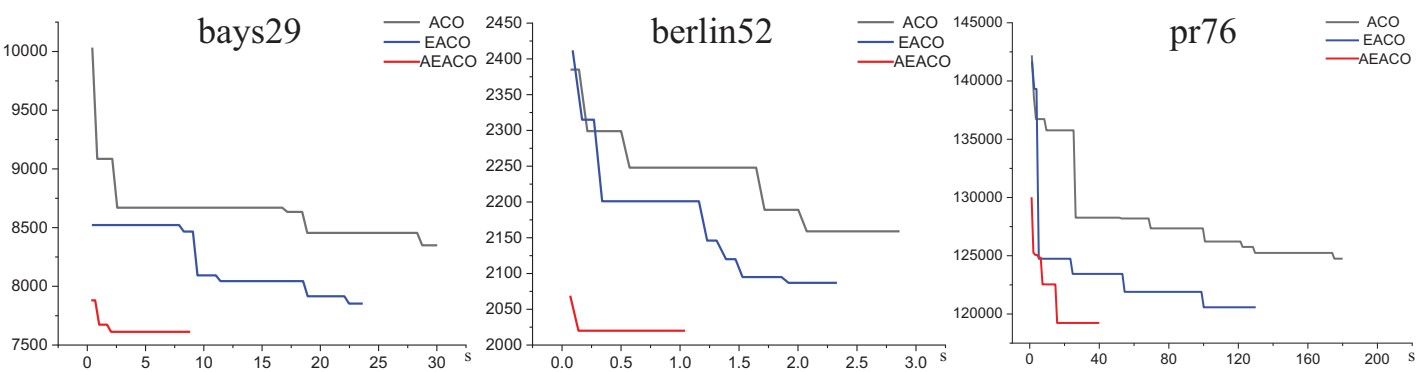

**Figure 1 Iterative trajectory of the best result of the ACO, EACO and AEACO solving small-scale instances.**

solution time for the algorithm to solve the TSP instance 30 times, *SD* is the error rate and calculated by Eq. (14), which is the difference between the optimal solution (denoted as Opt) obtained by the algorithm and the known optimal solution (denoted as KopS) of TSPLIB, $SD_{avg}$ is the average error of the solved result at the end of the every process after 30 runs, $SD_{best}$ is the error of the best solution after 30 runs and *Best* is best optimal solution after 30 runs.

$$SD = \left(\frac{Opt - KopS}{Kops}\right) \times 100 \tag{14}$$

Parameter setting of ACO: $\alpha = 1$, $\beta = 5$, $Q = 1$, $\rho = 1$, the number of ants $m$ is the number of cities.

Parameter setting of EACO: $\alpha = 1$, $\beta = 5$, $Q = 1$, $\rho = 1$, $e = 0.5$, the number of ants $m$ is the number of cities.

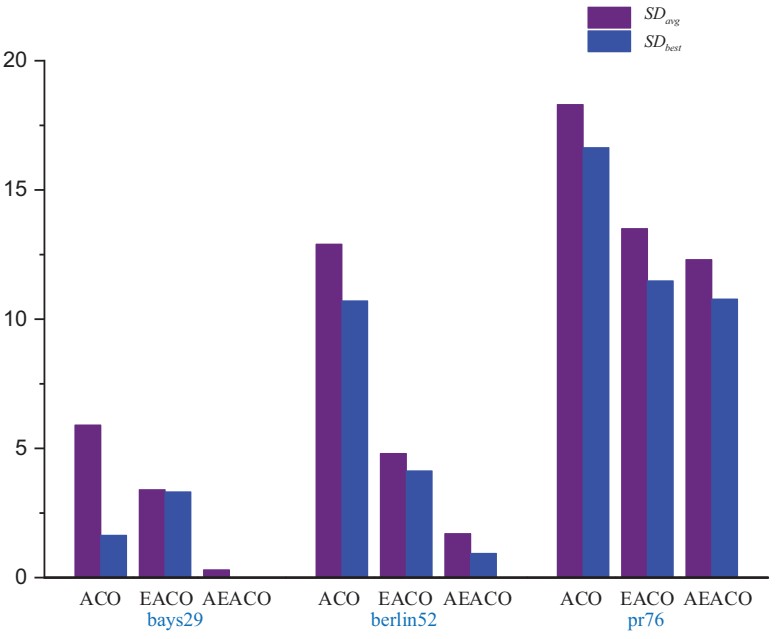

**Figure 2** Several errors in the results of the ACO, EACO and AEACO solving small-scale instances.

**Table 2** MSA-1 with different ratios for solving TSP instances.

| Instances ($K$) | 1:1:1 | | 2:1:1 | | 1:2:1 | | 1:1:2 | |
|---|---|---|---|---|---|---|---|---|
| | Agiter | $SD_{avg}$ | Agiter | $SD_{avg}$ | Agiter | $SD_{avg}$ | Agiter | $SD_{avg}$ |
| pr76 (512) | 346.70 | 7.40 | 366.77 | 6.01 | 377.57 | 7.48 | 335.83 | 6.08 |
| tsp225 (550) | 503.50 | 5.74 | 512.70 | 6.20 | 508.80 | 7.20 | 474.20 | 5.58 |
| pcb1173 (1,021) | 976.89 | 28.97 | 989.78 | 27.42 | 956.63 | 28.45 | 750.23 | 23.56 |

Parameter setting of AEACO: $\alpha = 7$, $\beta = 10$, $\rho = 0.1$, $Q = 1$, $e = 0.5$, $\lambda = 1.2$, the number of ants $m$ is the number of cities, the number of iterations of the algorithm is 0.5 times the number of cities.

From Fig. 1, the AEACO has demonstrated superior performance in terms of speed and accuracy compared to the ACO and EACO, it is evident that the AEACO converges to the optimal solution with minimal time and number of iterations, the convergence also speed is also remarkably fast as depicted in Fig. 2 and Table 2, it can be observed that the AEACO produces better results. Nevertheless, as the scale increases, the algorithm's accuracy diminishes, and the solution time considerably lengthens.

### Introduction and improvement of simulated annealing algorithm
#### Simulated annealing (SA)
According to *Metropolis et al. (1953)*, the basic idea behind solid annealing is to first slowly cool the solid after heating it to a specific temperature. When a solid is heated, its interior particles become disorganized as the temperature rises, and internal energy increases.

When a solid is cooled, on the other hand, its interior particle population becomes ordered as the temperature drops, and internal energy decreases when a particular equilibrium state is reached at each temperature. The SA replicates the steps involved in this principle, including the initial temperature setting, the initial solution, and the temperature decline.

The SA commences from an initial solution, denoted as $x$, and proceeds by perturbing $x$ according to predefined rules to generate a candidate solution, denoted as $y$. The Metropolis criterion is a fundamental acceptance rule used in the SA to determine whether a new solution should be accepted or rejected during the optimization process. The acceptance of $y$ is determined using the Metropolis criterion. If accepted, $y$ replaces $x$ as the new initial solution, from which further candidate solutions are generated. As the temperature decreases, the initial solution $x$ evolves iteratively. Eventually, this progressive evolution, driven by the decreasing temperature, leads the algorithm to converge towards the global optimal solution.

$$p = \begin{cases} 1 & E_y - E_x < 0 \\ e^{-\frac{E_y - E_x}{T_k}} & otherwise \end{cases} \tag{15}$$

$E_{old}$ and $E_{new}$ are the objective function value and $T_k$ is the current temperature.

The SA receives worse solutions with a certain probability. Therefore, the climbing ability is strong and it is not easy to fall into the local optimum, but the SA is slow to converge and has poor local search ability.

### Simulated annealing with multiple optimization seeking methods (MSA)

The SA is a powerful global optimization algorithm known for its excellent hill-climbing ability. However, the SA is heavily dependent on the initial temperature and a single optimization search method. As a result, the algorithm can fall into premature convergence and become trapped in local optimal solutions. Therefore, this article proposes two strategies to improve the deficiencies of the SA.

*Strategy one*

The traditional simulated annealing, which primarily employs perturbation operations to perturb the solution sequence, uses a random exchange of the positions of a specific pair of cities as its perturbation mechanism in the TSP instances, which is the primary cause of the algorithm's slow convergence. The perturbation mechanism, however, plays a critical role in the algorithm's superiority. As the perturbation approach for the SA process cannot be overly complicated, we will employ three conventional perturbation operations to change the old solution sequence.

(1) Swap method: randomly swap two positions in the solution sequence.

(2) Random insert method: randomly swap two adjacent positions in the solution sequence.

(3) 2-opt method: two positions in the solution sequence are randomly selected and arranged in reverse order from these two positions.

*Strategy two*

The traditional SA typically employ a sufficiently large initial temperature to enhance the search performance. However, there is no universally recommended method for determining the initial temperature. Selecting an inappropriate initial temperature not only results in wasted time but also hampers the effectiveness of the solution search. According to the method described in literature (*Lin, Kao & Hsu, 1993*), we employed a specific initial temperature approach tailored to the unique characteristics of our algorithm:

$$T = -\frac{E(X_{avg}) - E(X_{\min})}{a \ln p} \tag{16}$$

$E_{avg}$ and $E_{min}$ are the expected average and minimum values, respectively, of the objective function for N randomly selected feasible solutions within the solution space, $p$ is parameter [0,1], $a$ is a parameter value that mainly prevents the temperature starting point from being too high, resulting in slow convergence of the algorithm.

The MSA-1 is randomly selected to perturb the solution sequence by the swap method, random insertion method, and 2-opt method. The MSA-1 means that the MSA uses only strategy 1 and not strategy 2. In this article, three instances of pr76, tsp225, and pcb1173 are used to test the MSA-1 for four different proportions of perturbations of the swap method, random insertion method, and 2-opt method. The experimental results are shown in Table 2. In the table, *K* is the number of temperature changes, *Errors* is the average error of the solution after 30 experiments. Meanwhile, *Agiter* is the average of the number of iterations in which the optimal result emerges during the solution of the TSP instances after 30 times. $SD_{avg}$ is the average error rate of solving 30 times.

As shown in Table 2, when the *swap method*:*random insertion method*:*2−opt method* = 1:1:2, the solution effect is better. A larger proportion of the 2-opt method is beneficial for obtaining better solutions and reducing the number of iterations.

By utilizing the swap method, random insertion method, and 2-opt method to enhance perturbation in the SA. The perturbation capabilities of the SA can be improved, which can leading to more accurate solutions. Using strategy 2 to improve the initialization temperature of the SA has the advantage of avoiding too high or too low a temperature that would cause the algorithm to fall into a local optimal solution. In this study, comparison tests between the MSA-1 and the conventional the SA are carried out using the same initial solution sequence, initial temperature, termination temperature, cooling factor, and the maximum number of iterations. The MSA does not use the same initial temperature. A total of 30 tests are carried out, and the experimental results are displayed in Table 3 and Fig. 3.

The parameters were set as follows: initial temperature: 300, termination temperature 1, cooling factor: 0.998, and the maximum number of iterations: 100.

Experiments show that the solution accuracy of the MSA-1 is much higher than that of the SA in the same environment. The main reason is that Strategy 1 improves the exploration pattern of the SA. Comparing the MSA-1 and the MSA, MSA has strategy 2

**Table 3 Results of the SA, MSA-1 and MSA for solving different size instances.**

| Instances | Opt | Initial solution | Algorithm | Best | Worse | $SD_{avg}$ | Time (s) |
|---|---|---|---|---|---|---|---|
| pr76 | 108,159 | 144,831.45 | SA | 135,513.2 | 143,093.8 | 28.36 | 14.92 |
| | | | MSA-1 | 109,644.32 | 112,642.48 | 2.1 | 24.33 |
| | | | MSA | 108,202.16 | 114,270.43 | 1.12 | 66.39 |
| pr136 | 96,772 | 124,989.96 | SA | 113,226.76 | 118,201.66 | 19.2 | 23.13 |
| | | | MSA-1 | 99,787.90 | 104,539.30 | 5.8 | 44.38 |
| | | | MSA | 98,575.38 | 102,455.69 | 4.2 | 117.4 |
| lin318 | 42,029 | 53,884.25 | SA | 51,790.45 | 55,825.56 | 27.28 | 50.98 |
| | | | MSA-1 | 46,552.59 | 49,505.13 | 14.9 | 106.13 |
| | | | MSA | 44,424.19 | 45,297.42 | 7.37 | 286.13 |

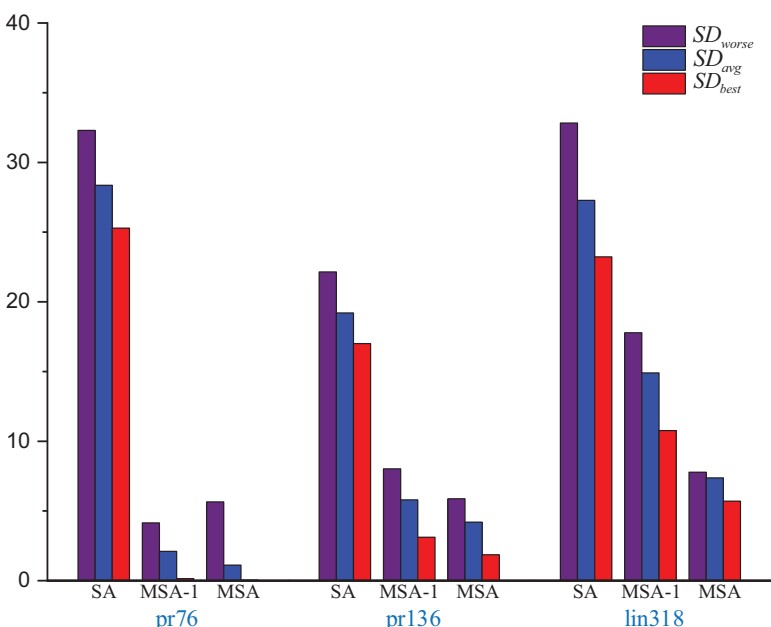

**Figure 3 Several errors in the results of the SA, MSA-1 and SA solving different size instances.**

which provides an effective initial temperature, leading to a high convergence accuracy of the MSA. But MSA of the average time is longer than the SA, resulting in the inability of the SA to improve the solution accuracy and reduce the solution time, mainly because the MSA is not sufficient in local search capability.

## A hybrid algorithm combining enhanced ant colony optimization and improved double simulated annealing *via* clustering (ACO-DSA)

After improving Ant Colony Optimization and simulated annealing, the AEACO helps the initial ant colony form a good search path and rewards the elite ant colony that finds the optimal path with extra pheromones. However, this algorithm cannot be used for the

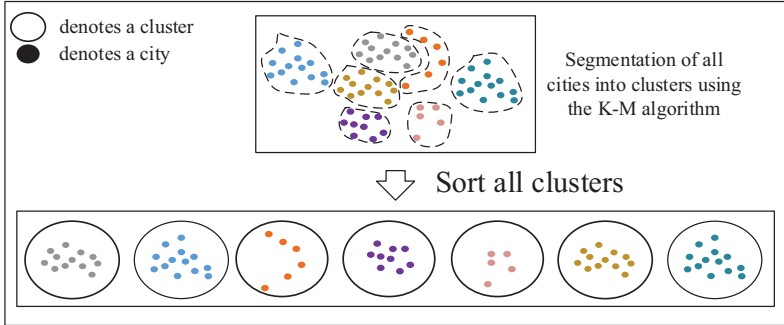

**Figure 4 Formation of small cluster classes and cluster sequences.**

larger TSP instances, mainly because the AEACO cannot get rid of the inherent defects of the ACO. While the enhanced simulated annealing increases the algorithm's perturbation of the solution sequence and improves its climbing ability and solution accuracy, it also leads to an increase in solution time. These two algorithms can work together as a complement. From an overall perspective, these two algorithms can complement each other, with the AEACO responsible for rapid search of small-scale TSP instances and the MSA responsible for the overall TSP instances to jump out the local optimal solution problem. Based on the above analysis, this article will use the clustering algorithm to combine the advantages of the AEACO and the MSA with each other.

## Steps of the algorithm

The algorithm in this article will be carried out in three processes.

Initialization process: Formation of small cluster classes and cluster sequences, intra-cluster optimization.

First annealing process: Optimizing the sequence of clusters and intra-cluster optimization.

Second annealing process: Global optimization.

### Formation of small cluster classes and cluster sequences

TSP instances can be fundamentally viewed as sorting problems, allowing large-scale TSP instances to be decomposed into multiple smaller ones. Solving the TSP involves determining the sequence in which the salesman selects the next city from the current city, typically focusing on cities around the current location. This approach is preferred over selecting cities across significant distances to maintain precision. To address this, as shown in Fig. 4, a clustering algorithm can be applied to partition all cities in the TSP instances into several small clusters, comprising cities that are in close proximity to each other. These clusters represent groups of closely located cities.

Steps for forming small cluster classes:

Step 1: Randomly select $k$ cities as *Medoids*.

Step 2: The remaining cities of the data are divided into cluster classes according to the principle of closest to the *Medoids*.

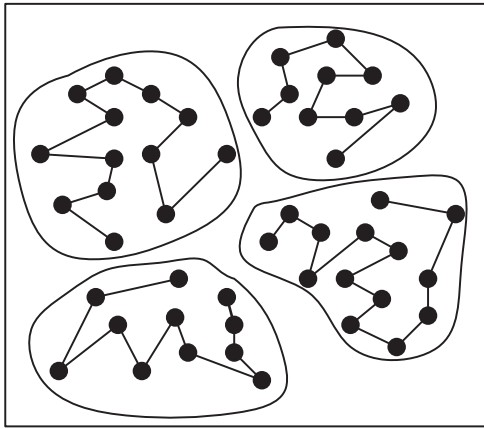

**Figure 5 Intra-cluster optimization.**

Step 3: Update: for each cluster formed by the assigned data points, select a new *Medoid* that minimizes the total dissimilarity or distance within that cluster. Iterate through each data point within the cluster and calculate the total dissimilarity as the sum of distances between the data point and all other points in the cluster. Choose the data point with the lowest total dissimilarity as the new *Medoid* for that cluster.

Step 4: Evaluate whether each cluster exceeds its maximum member capacity and consider reclassifying any surplus members from the exceeding clusters to other appropriate clusters.

Step 5: Repeat the process of 2–4 until all *Medoids* no longer change or the set maximum number of iterations has been reached.

Step 6: Forming cluster sequence: using the greedy algorithm and cluster centroids to sort the clusters.

### Intra-cluster optimization

The AEACO is an efficient algorithm for solving small-scale TSP instances with known starting city and ending city. Therefore, in this article's algorithm, the AEACO is used to find the best solution found for the TSP instances that are split into small-scale ones. As shown in Fig. 5, the AEACO is used to find the best cluster solutions inside each of the $m$ cluster classes as well as the best sequence of solutions.

Steps for finding the best solution founds and solution sequences within the cluster class:

Step 1: Parameter initialization: the number of $m$ ants (the number of cities), pheromone importance factor $\alpha$, heuristic function importance factor $\beta$, pheromone volatility factor $\rho$, total pheromone release $Q$, the maximal number of iterations *itermax* (0.5 times the number of cities), elite reward strategy value $e$, parameter $\lambda$.

Step 2: Initialization of pheromones: initialize the pheromone concentrations on each path according to Eq. (11).

Step 3: Construct the solution space:

Select the set of *cities set* from unvisited cities according to strategy two of AEACO (Eqs. (12) and (13)).

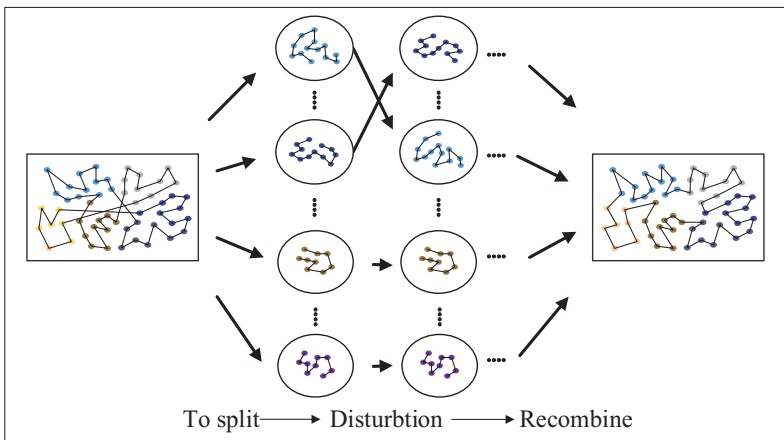

**Figure 6 Optimizing the sequence of clusters.**     

Each ant in the population selects the next city from *cities set* to move based on pheromone trails and heuristic information.

Repeat the process until all ants have visited all cities, ensuring that each city is visited exactly once.

Calculate the path length each ant and identify the ant with the best path (elite ant).

Step 4: Pheromone update: leave pheromones on each ant's passing edge and reward elite ants with a certain amount of extra pheromones on their passing edge.

Step 5: Termination condition: if *iter* < *itermax* (the current iteration number < the maximum iteration number), clear the path record table and return to Step 3; if *iter* >= *itermax* (the current iteration number >= the maximum iteration number), terminate the calculation and output the best solution found.

### Optimizing the sequence of clusters

The split into small-scale clusters needs to be recombined into a solution to the large-scale TSP instances, however, in the process of combination, a suitable sequence is needed for sorting. Therefore, the algorithm in this article uses the MSA to find the optimal ordering of this cluster. To obtain a better solution, the received sequence of clusters is first perturbed to produce a new sequence of clusters. Next, the optimal sequence of clusters is internally searched for each cluster in the new sequence of clusters, and the optimal solution and sequence of clusters are obtained by using the table of the nearest cities and the optimal solution of clusters obtained from the internal search of clusters. Lastly, the decision of whether to accept the new cluster sequence is made using the Metropolis criterion, as shown in Fig. 6.

### SE (Starting-Ending) strategy

As illustrated in the Fig. 7, the algorithm presented in this article primarily utilizes the clustering method to partition cities into distinct clusters. Subsequently, it aims to determine the optimal sequence for connecting these clusters. Hence, the most crucial consideration lies in selecting cities that are interconnected within their respective clusters. After the perturbation, it is necessary to determine the cities at the starting and ending

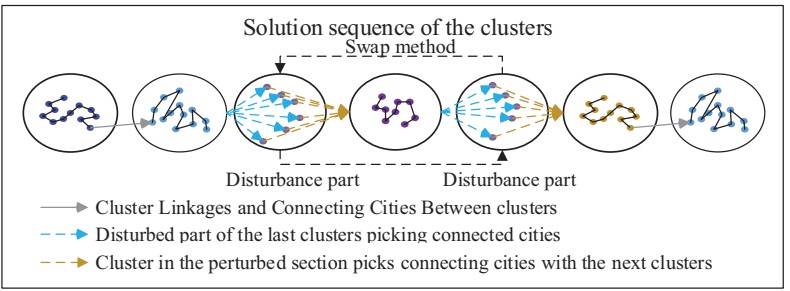

**Figure 7** **Schematic diagram of the SE strategy.**

points within the perturbed part based on the cluster. To achieve this, pheromones and distances from surrounding clusters to cities in the perturbed part are taken into account. Selection probabilities are established using these factors and are combined with the roulette pair method to select the starting city and ending city. The probability of selection of interconnected cities in the current cluster selection and other clusters is given by the formula:

$$\tilde{p}_{a\bar{k}}^{k} = \begin{cases} \dfrac{\sum_{b\in cluster_{\bar{k}}} [v_{ab}]^{\alpha_1} [\omega_{ab}]^{\beta_1}}{\sum_{r\in cluster_k} \sum_{a\in cluster_{\bar{k}}} [v_{rb}]^{\alpha_1} [\omega_{rb}]^{\beta_1}} & a \in cluster_k \\ 0 & otherwise \end{cases} \qquad (17)$$

Pheromone update of SE strategy:

$$\tau_{ij}(t+1) = (1-\ell_1)\tau_{ij}(t) + \frac{U_1}{L_1^{best}}(t) \qquad (18)$$

$a$ denotes a city in the cluster $k$ in the disturbed part, $v_{ab}$ denotes the heuristic factor between the city $a$ and the city $b$, $\omega_{ab}$ is the pheromone concentration between the city $i$ and the city $b$, $cluster_k$ denotes the cluster of the disturbed part, $cluster_{\bar{k}}$ denotes disturbed parts of the surrounding cluster, $\alpha_1$ denotes the pheromone factor, $\beta_1$ denotes the heuristic factor. $\ell_1$ is representative of the rate of volatilization, $U_1$ is a constant; $L_1^{best}$ is the current loop optimal solution of MSA.

Steps to optimize the sequence of clusters:

Step 1: Parameter initialization: temperature initialization $T_0$ operation with Eq. (16), the maximum number of iterations $L_0$, the termination temperature $T_{end}$, and the cooling factor $\rho_0(1 < \rho_0 < 0)$.

Step 2: Initialization of pheromones: set the same pheromone on each path.

Step 3: The initialization process of the cluster sequence: perform the internal cluster search for each cluster, and build up the optimal solution table of each cluster and the internal sequence table of each cluster according to the sequence of clusters.

Step 4: Perturbation of cluster sequence: perturbation of cluster sequence using swap method, random insertion method and 2-opt method with a probability of 1:1:2.

Step 5: Calculate the results after the perturbation:

Using SE strategy to find the starting city and the ending city of the perturbed part of the clusters.

Using an intra-cluster optimization search for the clusters in the perturbed part.

Update the optimal solution table and the internal sequence table of the clusters.

Step 6: The Metropolis criterion: use the Metropolis criterion to determine whether to receive a new cluster sequence.

Step 7: Determine if the maximum number of iterations has been reached: if so, exit the current loop, otherwise return to Step 3.

Step 8: Pheromone update: leave pheromones on each ant's passing edge and reward elite ants with a certain amount of extra pheromones on their passing edge.

Step 9: Cooling: $T_0 = \rho_0 T_0$.

Step 10: Stop condition: determine whether the termination temperature $T_{end}$ is reached, if $T_{now} = T_{end}$, the algorithm ends. Otherwise return to Step 3.

Step 11: Formation of a quality solution: connect all clusters.

### Global optimization

Since the clustering algorithm splits the TSP instances into several small TSP instances, mainly based on the nearest principle, resulting in the solution of this algorithm is not necessarily the optimal solution, due to the strong climbing ability of the MSA can jump out of the local optimal solution, so the global search for the sequence after experiencing the clustered sequence seeking is performed to increase the quality of the solution of the algorithm in this article.

Steps for global optimization search:

Step 1: Parameter initialization: initial temperature $T_1$ (Initial temperature calculated by combining Eq. (16) and historical data from Optimizing the sequence of clusters), the maximum number of iterations $L_1$, termination temperature $T_{END}$, cooling factor $\rho_1 (1 < \rho_1 < 0)$.

Step 2: Solution sequence perturbation: the solution sequence is perturbed using the swap method, random insertion method, and 2-opt method with a probability of 1:1:2.

Step 3: The Metropolis criterion: the Metropolis criterion is used to determine whether to receive new sequences and new solutions.

Step 4: Determine if the maximum number of iterations has been reached: if so, exit the current loop, otherwise return to Step 2.

Step 5: Cooling: $T_1 = \rho_1 T_1$.

Step 6: Stop condition: determine whether the termination temperature $T_{END}$ is reached, if $T_{NOW} = T_{END}$, the algorithm ends. Otherwise return to Step 2.

Step 7: Output the best solution found: output the best solution found and solution sequence.

### The flow of ACO-DSA

As shown in Fig. 8, we demonstrate the entire algorithmic flow of our algorithm.

**The process of initialization**

Step 1: The $n$ cities are clustered using the K-M algorithm.

Step 2: Using the greedy algorithm and cluster centroids, create a cluster sequence.

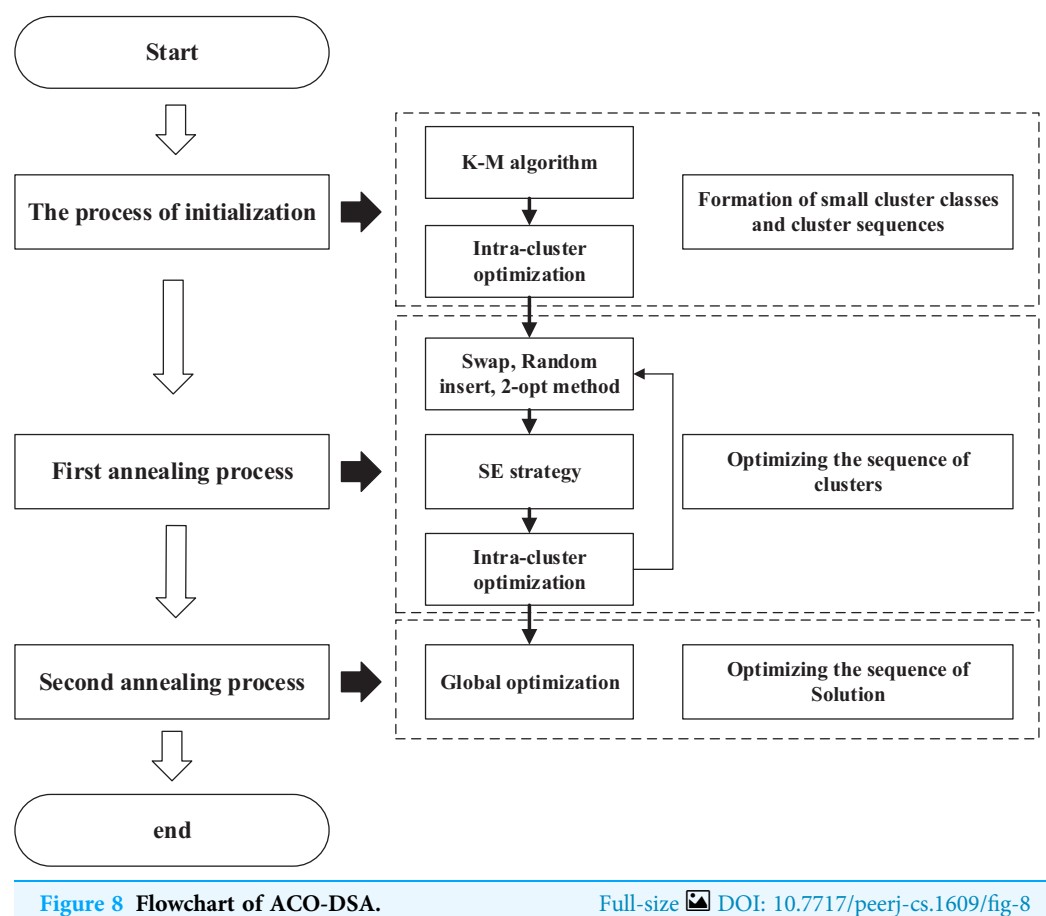

**Figure 8 Flowchart of ACO-DSA.**

Step 3: The starting and ending cities of each cluster are selected using the SE strategy. The AEACO is then employed to find the best solution found and sort each cluster based on the cluster class order. Finally, the optimal solution table and the internal sequential table are constructed.

**First annealing process**

Step 4: Initial temperature, termination temperature, number of iterations, and cooling factor of intra-cluster optimization are initialized for the operation.

Step 5: Perturb the cluster sequence and perform intra-cluster optimization search for the cities in the perturbed part of the cluster to calculate the solution value of the perturbed part.

Step 6: If the new solution sequence is accepted, determine using the Metropolis acceptance criterion: whether to update the optimal solution table for each cluster and the internal sequence table for each cluster; if not, do not update these two tables.

Step 7: Judge whether the number of iterations is reached, if not, return to Step 5, otherwise, execute the next step.

Step 8: Judge whether the termination temperature is reached, if not, return to Step 5, otherwise find out the solution sequence and solution based on the optimal solution table of each cluster and the internal sequence table of each cluster, and execute the next step.

**Second annealing process**

Step 9: Initial temperature, termination temperature, number of iterations, and cooling factor of global optimization are initialized for the operation.

Step 10: Perturb the solution sequence and calculate the new solution.

Step 11: Use the Metropolis acceptance criterion to judge whether the new solution sequence is received, if so, update the solution sequence; otherwise, do not update the solution sequence.

Step 12: Judge whether the number of iterations is reached, if not, return to Step 10, otherwise, execute the next step.

Step 13: Judge whether the termination temperature is reached, if not, return to Step 10, otherwise, output the solution sequence and the best solution found.

# EXPERIMENT AND RESULT ANALYSIS

To test the effectiveness of the ACO-DSA, experiments will be conducted using TSP instance of different sizes from the TSPLIB database, which are arranged as follows:

(1) The ACO-DSA is used to find the optimal solution for different TSP instance sizes.

(2) The effect of varying clusters of clusters on the first annealing at the same TSP scale and temperature control.

(3) The ACO-DSA is compared with other algorithms.

This article experiments in the following environment: Python 3.8, Win11 operating system, Intel(R) Core(TM) i7-8550U processor, 8 GB RAM.

## Testing the solution effect of the ACO-DSA

To verify the operational effectiveness of the ACO-DSA, 30 tests are run on TSP instances of different sizes. In particular, Table 4 shows the relevant parameter settings of the algorithm in this article for each instance. In the table, Size represents the number of clusters into which the TSP instance is divided. Meanwhile, parameter settings of AEACO and SE strategy are: $\alpha = 7, \beta = 10, \rho = 0.1, Q = 1, e = 0.5, \lambda = 1.2$, The number of ants is the number of cities, The number of iterations of the algorithm is 0.5 times the number of cities, $\alpha_1 = 7, \beta_1 = 10$. Table 5 shows the relevant results of each instance in each process and the average of the sum of the optimal solutions that emerge after 30 times of conducting and the total time required for each experiment. *Time* is the solution time of the process, *SD* is the error rate and calculated by Eq. (14), $SD_{avg}$ is the average error of the solved result at the end of the every process after 30 runs, $SD_{best}$ is the error of the best result after 30 runs and *Best* is the best result after 30 runs.

According to experiments, when using the ACO-DSA to solve instances of various sizes, the solution accuracy increases at the end of each process and converges to a specific range of accuracy. Figure 9 shows some examples of optimal paths obtained by the ACO-DSA, every result was excellent.

## Testing the impact of cluster size

Decomposing the instance into numerous small clusters and effectively ranking each cluster is the primary priority of the first annealing process of the ACO-DSA. The effect of the number of clusters on the time and accuracy of the ACO-DSA is investigated by

**Table 4 Parameter settings of the ACO-DSA for different TSP instances.**

| Instances | Size | First annealing process | | | | | Second annealing process | | | | |
|---|---|---|---|---|---|---|---|---|---|---|---|
| | | $a_0$ | $p_0$ | $\rho_0$ | $L_0$ | $T_{end}$ | $a_1$ | $p_1$ | $\rho_1$ | $L_1$ | $T_{END}$ |
| eil51 | 10 | 2 | 0.01 | 0.998 | 1 | 0.05 | 0.01 | 2 | 0.998 | 100 | 0.1 |
| berlin52 | 10 | 2 | 0.01 | 0.998 | 1 | 2 | 0.01 | 2 | 0.998 | 100 | 2 |
| st70 | 10 | 2 | 0.01 | 0.998 | 1 | 1 | 0.01 | 2 | 0.998 | 100 | 0.5 |
| eil76 | 10 | 2 | 0.01 | 0.998 | 1 | 0.07 | 0.01 | 2 | 0.998 | 150 | 0.002 |
| pr76 | 10 | 2 | 0.01 | 0.998 | 1 | 50 | 0.01 | 5 | 0.998 | 150 | 50 |
| kroA100 | 15 | 2 | 0.01 | 0.998 | 1 | 7 | 0.01 | 5 | 0.998 | 200 | 5 |
| eil101 | 10 | 2 | 0.01 | 0.998 | 1 | 0.1 | 0.01 | 2 | 0.998 | 200 | 0.1 |
| pr107 | 10 | 2 | 0.01 | 0.998 | 1 | 20 | 0.01 | 2 | 0.998 | 200 | 5 |
| bier127 | 15 | 2 | 0.01 | 0.998 | 1 | 40 | 0.01 | 5 | 0.998 | 200 | 1 |
| ch130 | 15 | 2 | 0.01 | 0.998 | 1 | 1.2 | 0.01 | 2 | 0.998 | 200 | 1.2 |
| pr136 | 15 | 2 | 0.01 | 0.998 | 1 | 20 | 0.01 | 2 | 0.998 | 200 | 5 |
| ch150 | 20 | 2 | 0.01 | 0.998 | 1 | 1 | 0.01 | 4 | 0.998 | 250 | 1 |
| kroA200 | 20 | 2 | 0.01 | 0.998 | 1 | 0.1 | 0.01 | 4 | 0.998 | 250 | 1 |
| tsp225 | 20 | 2 | 0.01 | 0.998 | 1 | 0.5 | 0.01 | 5 | 0.998 | 300 | 1 |
| pr299 | 30 | 2 | 0.01 | 0.998 | 5 | 10 | 0.01 | 20 | 0.998 | 300 | 1 |
| lin318 | 40 | 2 | 0.01 | 0.998 | 1 | 7 | 0.01 | 5 | 0.998 | 400 | 1 |
| pr439 | 50 | 2 | 0.01 | 0.998 | 1 | 20 | 0.01 | 20 | 0.998 | 500 | 0.5 |
| rat575 | 80 | 2 | 0.01 | 0.998 | 50 | 10 | 0.01 | 20 | 0.998 | 800 | 0.01 |
| p654 | 100 | 2 | 0.01 | 0.998 | 50 | 5 | 0.01 | 5 | 0.998 | 500 | 0.1 |
| rat783 | 150 | 2 | 0.01 | 0.998 | 1 | 1 | 0.01 | 1 | 0.998 | 800 | 0.001 |
| vm1084 | 150 | 2 | 0.01 | 0.998 | 50 | 10 | 0.01 | 20 | 0.998 | 1,000 | 1 |
| d2103 | 400 | 2 | 0.01 | 0.998 | 200 | 10 | 0.01 | 20 | 0.998 | 1,500 | 1 |

varying the number of clusters and the number of iterations while maintaining the same termination temperature and cooling factor. As depicted in Table 6, we illustrate the influence of the number of clusters and the number of iterations on the initial stage of the proposed algorithm using the pr299 instance. The horizontal axis of the table represents the number of cities in the cluster (5, 10, 15, and 20), while the vertical axis represents the number of iterations $L_0$ (1, 5, 10, 15, 20, and 25).

The formula for calculating the number of cities in the cluster is shown below:

$$cities = \frac{\text{Number of all cities}}{\text{Number of cluster classes}} \qquad (19)$$

Parameter settings for the pr299 instances: termination temperature: 10, cooling factor: 0.998, $a_0 = 2$, $p_0 = 0.01$ (*Error* indicates the average error rate of solving 10 times, *Time* represents the solution time for the first annealing process of the ACO-DSA to solve the TSP instance 10 times).

The results indicate that increasing the number of cities in the cluster leads to longer optimization times. However, it also achieves better optimization results for the same case.

Table 5 Results of the operation of the ACO-DSA clustering for different TSP instances.

| Instances | Opt | Initialization process | | First annealing process | | Second annealing process | | $SD_{best}$ | Best |
|---|---|---|---|---|---|---|---|---|---|
| | | Time (s) | $SD_{avg}$ | Time (s) | $SD_{avg}$ | Time (s) | $SD_{avg}$ | | |
| eil51 | 426 | 0.69 | 10.40 | 5.75 | 5.96 | 4.15 | 3.28 | 0.67 | 428.9 |
| berlin52 | 7,542 | 0.62 | 29.86 | 4.28 | 6.66 | 6.27 | 1.94 | 0.03 | 7,544.4 |
| st70 | 675 | 0.80 | 11.24 | 8.13 | 6.04 | 0.83 | 3.99 | 0.31 | 677.1 |
| eil76 | 538 | 1.07 | 15.42 | 20.69 | 9.57 | 18.14 | 3.99 | 1.92 | 548.3 |
| pr76 | 108,159 | 1.11 | 15.40 | 20.60 | 5.99 | 9.36 | 2.74 | 0.00 | 108,159.4 |
| kroA100 | 21,282 | 0.78 | 22.82 | 17.47 | 6.80 | 18.99 | 2.32 | 0.47 | 21,381.8 |
| eil101 | 629 | 0.79 | 17.99 | 65.50 | 9.37 | 17.01 | 5.75 | 4.30 | 656.0 |
| pr107 | 44,303 | 2.17 | 9.29 | 16.36 | 3.66 | 61.53 | 0.70 | 0.00 | 44,301.7 |
| bier127 | 118,282 | 8.97 | 16.89 | 76.78 | 8.98 | 80.85 | 3.85 | 0.36 | 118,703.6 |
| ch130 | 6,110 | 4.47 | 18.88 | 60.84 | 7.81 | 27.09 | 4.13 | 0.12 | 6,117.5 |
| pr136 | 96,772 | 6.88 | 10.55 | 87.83 | 4.77 | 24.29 | 2.74 | 0.62 | 97,367.8 |
| ch150 | 6,528 | 6.74 | 17.05 | 71.68 | 9.38 | 25.35 | 4.81 | 0.45 | 6,557.3 |
| kroA200 | 29,368 | 12.15 | 18.80 | 101.98 | 10.23 | 111.51 | 5.84 | 3.61 | 30,428.7 |
| tsp225 | 3,916 | 3.40 | 17.74 | 96.05 | 7.99 | 72.54 | 3.94 | 1.40 | 3,970.9 |
| pr299 | 48,191 | 8.93 | 25.74 | 706.82 | 8.87 | 185.36 | 3.84 | 0.02 | 48,200.6 |
| lin318 | 42,029 | 9.65 | 29.87 | 164.10 | 14.22 | 195.36 | 6.02 | 3.93 | 43,680.7 |
| pr439 | 107,217 | 11.73 | 29.37 | 478.63 | 12.66 | 369.67 | 7.50 | 4.24 | 111,763.1 |
| rat575 | 6,773 | 28.31 | 28.91 | 239.53 | 15.25 | 646.80 | 8.63 | 3.90 | 7,036.8 |
| p654 | 34,643 | 29.31 | 32.91 | 134.77 | 16.25 | 746.80 | 6.12 | 3.22 | 35,757.4 |
| rat783 | 8,806 | 25.87 | 33.74 | 268.14 | 12.81 | 876.58 | 10.71 | 5.01 | 9,247.5 |
| vm1084 | 239,297 | 30.80 | 38.74 | 768.16 | 15.81 | 1,176.55 | 11.68 | 8.36 | 259,303.1 |
| d2103 | 80,450 | 56.65 | 40.76 | 2,000.87 | 20.76 | 2,098.76 | 13.87 | 10.28 | 88,722.59 |

Similarly, when maintaining the same number of cities in the cluster, a higher number of iterations results in superior optimization outcomes, albeit with a significant increase in processing time. Specifically, when using 20 cities in the cluster and 25 iterations, the solution accuracy reaches its peak. Nevertheless, it is not recommended to choose a configuration with a large number of cities in the cluster and a high number of iterations due to the considerable time consumption.

## Comparison with other algorithms

Table 7 displays the experimental results concerning the algorithm proposed in this article and several traditional algorithms, namely Ant Colony optimization (ACO), simulated annealing (SA), genetic algorithm (GA), and particle swarm optimization (PSO), after conducting 30 tests. The experimental records indicate that as the size of the instance increases, both ACO and PSO exhibit a deteriorating trend in solution accuracy and solution time performance. However, GA and PSO tend to converge to a certain level of accuracy. Remarkably, when compared to the traditional algorithms, the algorithm proposed in this article demonstrates superior performance. In terms of the average

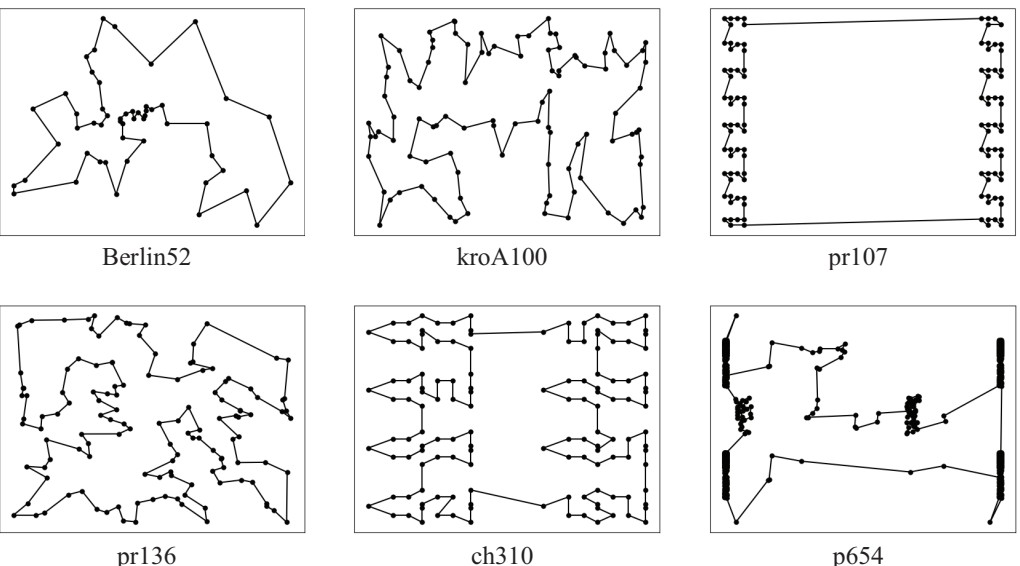

**Figure 9** Some examples of optimal paths obtained by the ACO-DSA.

**Table 6 Results of testing the number of different clusters and different iterations of pr299 instance.**

| Opt | $T_0$ | $L_0$ | 5 | | 10 | | 15 | | 20 | |
|---|---|---|---|---|---|---|---|---|---|---|
| | | | Time (s) | Error | Time (s) | Error | Time (s) | Error | Time (s) | Error |
| 48,191 | 10 | 1 | 15.73 | 16.93 | 130.18 | 11.03 | 338.44 | 8.90 | 589.09 | 7.33 |
| | | 5 | 68.52 | 12.31 | 706.82 | 8.87 | 1,076.87 | 8.45 | 2,459.12 | 6.98 |
| | | 10 | 147.35 | 9.76 | 1,201.82 | 8.66 | 1,809.76 | 8.19 | 4,989.82 | 6.89 |
| | | 15 | 210.97 | 9.24 | 1,765.56 | 8.39 | 2,982.89 | 7.83 | 6,878.32 | 6.70 |
| | | 20 | 280.64 | 8.20 | 2,105.67 | 7.88 | 3,650.88 | 7.40 | 8,887.66 | 6.38 |
| | | 25 | 330.45 | 7.52 | 2,650.84 | 7.32 | 4,205.59 | 7.20 | 10,965.88 | 5.88 |

optimal solution error rate across all instances, the algorithm proposed in this article achieves a value of 1.49, while ACO, SA, GA, and PSO attain values of 29.39, 5.59, 2.84, and 11.16, respectively. Additionally, considering the average value of the average error rate across all instances, this article's algorithm exhibits a value of 3.98, while ACO, SA, GA, and PSO exhibit values of 35.62, 9.30, 4.97, and 13.44, respectively. Moreover, the algorithm proposed in this article also demonstrates advantages in terms of solution time.

Figure 10 shows the analyzed graphs of the results of this article's algorithm and the ACO, SA, GA, and PSO. The line graphs show the error rates of the optimal solutions obtained by the SA, GA, PSO, and this article's algorithm after 30 solving of 10 TSP instances, and it can be seen that this article's algorithm achieves the best optimal solution. The bar chart shows the comparison of the two error rates of the ACO, SA, GA, and PSO.

Table 8 displays the experimental results concerning the algorithm proposed in this article and several other literature algorithms, the design idea of the ACO-ABC (*Gunduz, Kiran & Ozceylan, 2015*) in the literature is the same as the algorithm in this article, both of

**Table 7 Comparison results of the ACO-DSA with other traditional algorithms for different TSP instances.**

| | | ACO | | | SA | | | GA | | | PSO | | | This article | | |
|---|---|---|---|---|---|---|---|---|---|---|---|---|---|---|---|---|
| Instances | Opt | $SD_{best}$ | $SD_{avg}$ | Time | $SD_{best}$ | $SD_{avg}$ | Time | $SD_{best}$ | $SD_{avg}$ | Time | $SD_{best}$ | $SD_{avg}$ | Time | $SD_{best}$ | $SD_{avg}$ | Time |
| berlin52 | 7,542 | 10.71 | 12.90 | 27.81 | 0.03 | 5.81 | 46.51 | 0.03 | 6.14 | 18.10 | 1.00 | 6.74 | 12.15 | 0.03 | 1.94 | 11.17 |
| pr76 | 108,159 | 16.63 | 18.30 | 170.91 | 21.57 | 23.89 | 70.17 | 1.74 | 3.58 | 56.87 | 12.04 | 16.89 | 32.93 | 0.00 | 2.74 | 31.07 |
| kroA100 | 21,282 | 62.45 | 81.25 | 319.34 | 2.12 | 4.46 | 112.68 | 0.77 | 2.19 | 122.04 | 11.02 | 14.38 | 66.55 | 0.47 | 2.32 | 37.24 |
| eil101 | 629 | 19.66 | 23.96 | 350.35 | 6.68 | 10.53 | 124.45 | 5.21 | 6.88 | 137.02 | 14.61 | 16.54 | 57.99 | 4.30 | 5.75 | 83.30 |
| pr107 | 44,303 | 7.24 | 9.72 | 450.34 | 3.16 | 9.55 | 131.96 | 0.54 | 0.95 | 156.80 | 1.84 | 2.80 | 62.44 | 0.00 | 0.70 | 80.06 |
| ch130 | 6,110 | 19.35 | 20.48 | 875.44 | 2.73 | 5.71 | 236.96 | 3.25 | 5.78 | 246.96 | 12.40 | 16.06 | 67.69 | 0.12 | 4.13 | 92.40 |
| ch150 | 6,528 | 21.76 | 24.30 | 1,055.32 | 3.23 | 7.50 | 268.59 | 1.14 | 2.42 | 238.59 | 7.33 | 8.27 | 92.79 | 0.45 | 4.81 | 103.77 |
| tsp225 | 3,916 | 26.88 | 28.84 | 1,972.25 | 3.96 | 6.77 | 451.99 | 2.50 | 4.57 | 421.99 | 15.66 | 17.86 | 208.85 | 1.40 | 3.94 | 171.99 |
| lin318 | 42,029 | 51.09 | 67.73 | 2,501.34 | 5.48 | 9.10 | 788.48 | 5.66 | 6.26 | 2,725.49 | 15.38 | 16.57 | 567.96 | 3.93 | 6.02 | 369.11 |
| pr439 | 107,217 | 58.09 | 68.73 | 3,450.45 | 6.84 | 9.67 | 1,521.78 | 7.94 | 10.88 | 3,721.78 | 20.29 | 18.27 | 1,087.44 | 4.24 | 7.50 | 860.03 |
| Average | | 29.39 | 35.62 | 1,117.36 | 5.58 | 9.30 | 375.36 | 2.88 | 4.97 | 784.56 | 11.16 | 13.44 | 225.68 | 1.49 | 3.98 | 184.01 |

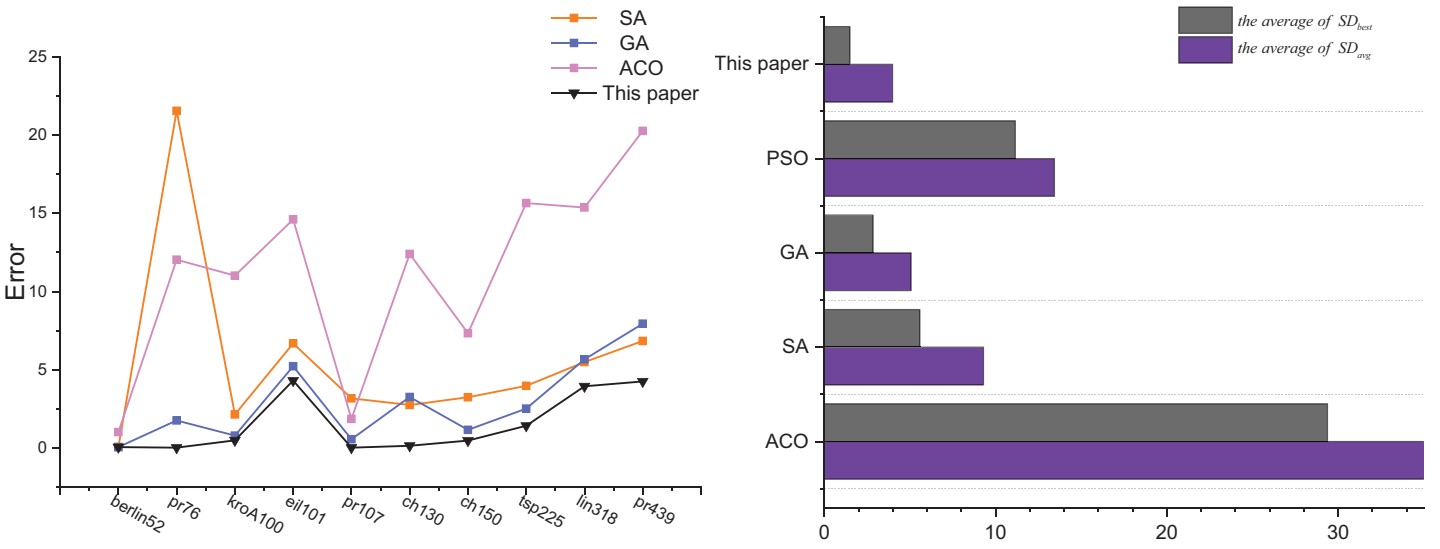

**Figure 10 Comparison results the ACO-DSA with other traditional algorithms in terms of solution accuracy for different TSP instances.**

them use the ACO to obtain the initial solution, and finally use other optimization methods to improve the initial solution. The IGSSA (*He, Wu & Xu, 2018*) and the ACO-PSO (*Qian & Su, 2018*) are respectively improvements to the Ant Colony Optimization and simulated annealing (Note: where '-' indicates that they are not mentioned in their literature).

Figure 11 shows the comparison of this article's algorithm with other algorithms in the literature. The average of $SD_{avg}$ and the average of $SD_{best}$ are the optimal solution error rate and the average error rate for all the TSP instances solution results mentioned in other literature and compared with the results of this article's algorithm. From the analysis of

**Table 8 Comparison results of the ACO-DSA with other methods in the literature for different TSP instances.**

| | | ACO-ABC | | | IGSSA | | | ACO-PSO | | | This article | | |
|---|---|---|---|---|---|---|---|---|---|---|---|---|---|
| Instances | Opt | Best | $SD_{best}$ | $SD_{avg}$ | Best | $SD_{best}$ | $SD_{avg}$ | Best | $SD_{best}$ | $SD_{avg}$ | Best | $SD_{best}$ | $SD_{avg}$ |
| berlin52 | 7,542 | 7,544.4 | 0.03 | 0.03 | 7,544.4 | 0.03 | 0.60 | 7,663.6 | 1.61 | 2.76 | 7,544.4 | 0.03 | 1.94 |
| bier127 | 118,282 | — | — | — | — | — | — | 124,842.7 | 5.55 | 6.37 | 118,703.6 | 0.36 | 3.85 |
| ch130 | 6,110 | — | — | — | — | — | — | 6,473.5 | 5.95 | 6.80 | 6,117.5 | 0.12 | 4.13 |
| ch150 | 6,528 | 6,641.7 | 1.74 | 2.28 | — | — | — | 6,852.1 | 4.96 | 5.49 | 6,557.3 | 0.45 | 4.81 |
| eil101 | 629 | 672.7 | 6.95 | 8.65 | — | — | — | 700.7 | 11.40 | 12.35 | 656.0 | 4.30 | 5.75 |
| eil51 | 426 | 431.7 | 1.35 | 4.08 | 428.9 | 0.67 | 1.17 | 448.6 | 5.29 | 6.31 | 428.9 | 0.67 | 3.28 |
| eil76 | 538 | 565.5 | 2.42 | 3.71 | 544.4 | 1.18 | 1.80 | 568.0 | 5.58 | 6.01 | 548.3 | 1.92 | 3.99 |
| kroA100 | 21,282 | 22,122.8 | 3.95 | 5.42 | — | — | — | 22,387.6 | 5.20 | 8.53 | 21,381.8 | 0.47 | 2.32 |
| pr107 | 44,303 | — | — | — | 44,301.7 | 0.00 | 0.21 | 46,249.4 | 4.39 | 5.08 | 44,301.7 | 0.00 | 0.70 |
| pr136 | 96,772 | — | — | — | 98,169.3 | 1.44 | 2.49 | — | — | — | 97,367.8 | 0.62 | 2.74 |
| pr76 | 108,159 | 113,798.6 | 5.21 | 6.39 | — | — | — | — | — | — | 108,159.4 | 0.00 | 2.74 |
| st70 | 675 | 687.2 | 1.81 | 3.79 | 677.1 | 0.31 | 0.96 | — | — | — | 677.1 | 0.31 | 3.99 |
| tsp225 | 3,916 | 4,090.5 | 4.46 | 6.18 | — | — | — | — | — | — | 3,970.9 | 1.40 | 3.94 |

Fig. 11, it is obtained that this article's algorithm is better than ACO-ABC and ACO-PSO, and is slightly worse than IGSAA in terms of the average error rate, but the IGSAA can only be used in small-scale TSP instances and does not apply to large-scale TSP instances.

## APPLICATION OF THE ACO-DSA ON LOGISTICS

Figure 12 below illustrates a map containing 44 cities that a company needs to travel to for transporting supplies. The journey starts and ends in Guizhou, forming a closed-loop route. The primary objective of the solution is to determine the optimized route and the corresponding distance to be traveled, encompassing all 44 cities. In the actual distribution process, there are the following situations: (1) the geographic locations of the distributing center and customer points are known; (2) the shortest path between each customer point and the distributing center is known; (3) the path of distribution starts from the distributing center and needs to return to the distributing center after completing all deliveries, forming a closed-loop distribution route; (4) each customer points can only be reached once; (5) the effect of road factors on the vehicle is not considered. The following Fig. 12 shows the customer points that a logistics company needs to distribute and the optimal path obtained by using the ACO-DSA (The white car represents the distributing center).

### Results of the simulation

The path planning for this company's problem is carried out by the ACO, EACO, AEACO, SA, MSA, and this article's algorithm based on the customer points indicated above. Table 9 records the minimum, maximum, and average values of this outcome. In the

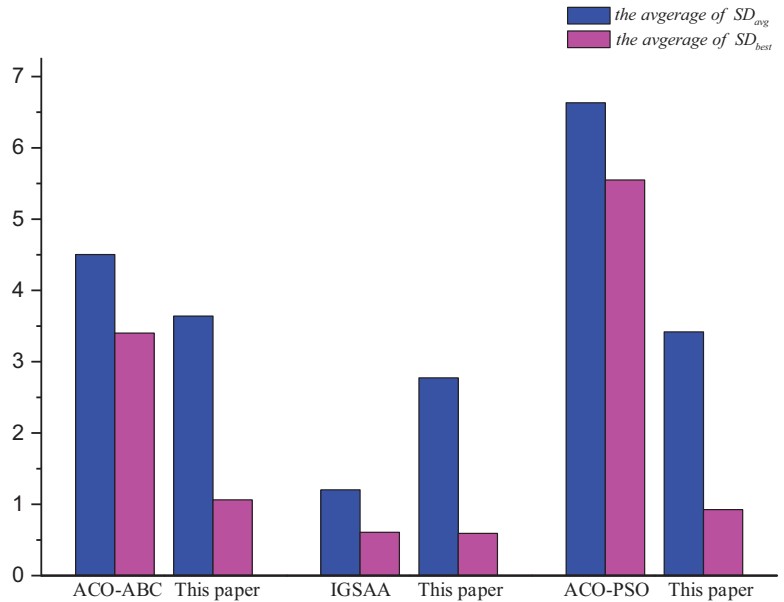

**Figure 11 Comparison results ACO-DSA with other methods in the literature in terms of solution accuracy for different TSP instances.**

**Figure 12 Results of ACO-DSA for the company.**

**Table 9 Results of the company's logistics transportation path solution.**

| Algorithm | Time | Min/km | Max/km | Mean/km |
|---|---|---|---|---|
| ACO | 30.89 | 14,464.58 | 15,154.65 | 14,819.02 |
| EACO | 28.66 | 14,293.87 | 14,573.39 | 14,480.73 |
| AEACO | 12.01 | 14,007.93 | 14,840.03 | 14,522.12 |
| SA | 14.31 | 14,483.91 | 14,789.76 | 14,590.89 |
| MSA | 24.57 | 13,524.50 | 13,657.97 | 13,559.09 |
| This article | 12.57 | 13,524.50 | 13,623.39 | 13,553.25 |

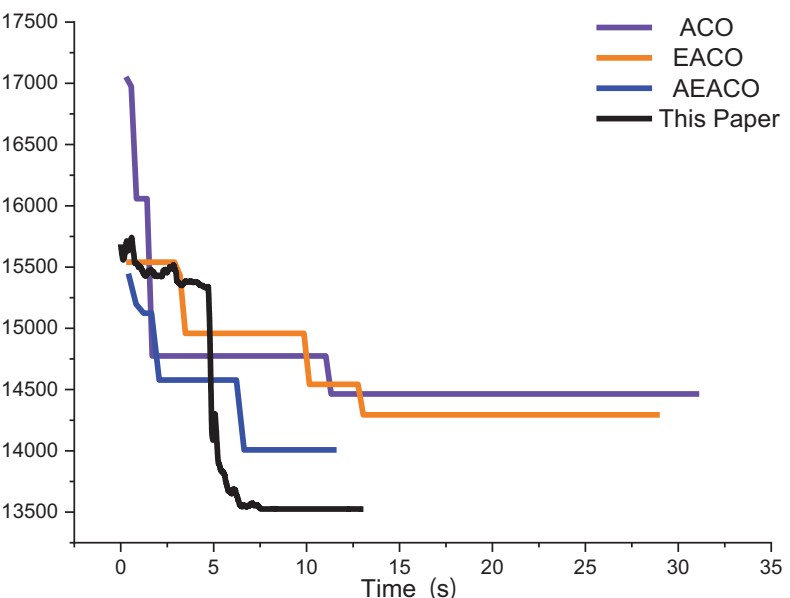

**Figure 13 Iteration results of the ACO, EACO, AEACO and ACO-DSA.**

simulation results, both MSA and this article's algorithm find the minimum optimized path length of 13,524.50 km for this company, but this article's algorithm has the shortest solution time and the highest solution quality. Meanwhile, from the average value, the solution effect of this algorithm is the most stable.

The approximate iterative trajectories of the SA, MSA, and the algorithm in this article are shown for better comparison. Figure 13 shows the iterative results of the ACO, EACO, AEACO and this article's algorithm for solving the path planning of this company. Compared with the ACO, EACO and AEACO, this article's algorithm converges faster, has better initialization results and takes less time to solve. Figure 14 shows the comparison of the optimization effect of SA, MSA and the algorithm in this article. From the results, the convergence speed and optimization effect of this article's algorithm are much better than SA and MSA.

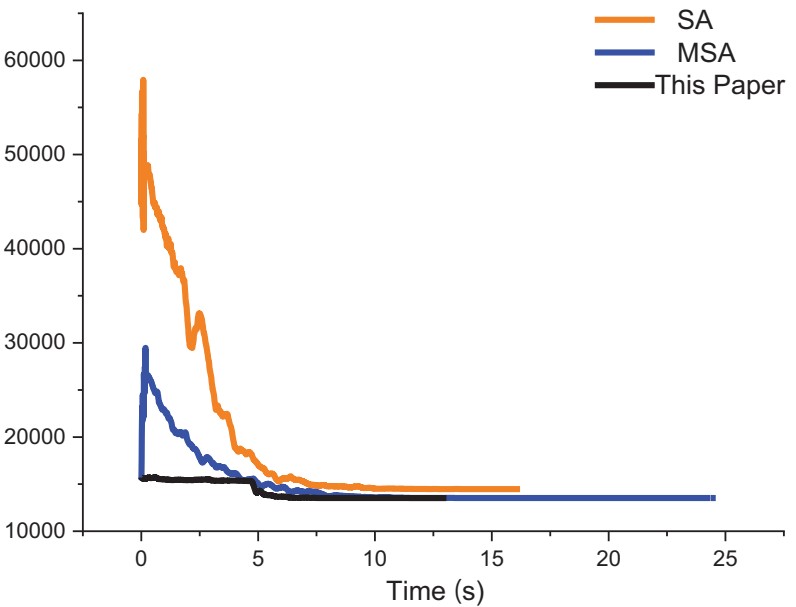

**Figure 14 Iteration results of the SA, MSA and ACO-DSA.**

## CONCLUSION

The ACO-DSA can achieve good results in solving TSP instances. All of them can converge to good accuracy and have some robustness when facing small to medium TSP instance sizes. The ACO-DSA can give a good result when faced with large scales, but the time consumed can be exceptionally long. In addition, both the algorithm in this article and the traditional algorithm are applied to logistics, and the algorithm in this article performs optimally.

As shown in Tables 4 and 5, The deficiency in the article is that it requires more parameters to be adjusted, and the main parameters are determined to be taken within a certain range after several experiments, which also increases the uncertainty of operation to a certain extent and there is room for improvement. At the same time, it takes more time to solve the TSP in the face of an ultra-large scale. In summary, the algorithm of this article will be improved in the future to further reduce its solve time. In the future, our focus will be on enhancing the AEACO, primarily by introducing changes that accommodate increased solution time as the size of the TSP instances increases. Secondly, the main improvement is to improve the number of iterations of the ACO-DSA, which leads to long solving time of the ACO-DSA due to the high number of iterations.

### Funding

This work was supported by the Classified Development Project of Beijing Universities (Grant No. 71R2211001). The funders had no role in study design, data collection and analysis, decision to publish, or preparation of the manuscript.

## Grant Disclosures

The following grant information was disclosed by the authors:
Classified Development Project of Beijing Universities: 71R2211001.

## Competing Interests

The authors declare that they have no competing interests.

## Author Contributions

- Tan Hao conceived and designed the experiments, performed the experiments, analyzed the data, performed the computation work, prepared figures and/or tables, authored or reviewed drafts of the article, and approved the final draft.
- Wu Yingnian conceived and designed the experiments, authored or reviewed drafts of the article, and approved the final draft.
- Zhang Jiaxing conceived and designed the experiments, performed the experiments, analyzed the data, prepared figures and/or tables, and approved the final draft.
- Zhang Jing conceived and designed the experiments, performed the experiments, analyzed the data, performed the computation work, prepared figures and/or tables, and approved the final draft.

## Data Availability

The raw data and code is available in the Supplemental Files.

## Supplemental Information

Supplemental information for this article can be found online at http://dx.doi.org/10.7717/peerj-cs.1609#supplemental-information.

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
