# Peer review of "Study on a hybrid algorithm combining enhanced ant colony optimization and double improved simulated annealing via clustering in the Traveling Salesman Problem (TSP)"

_PeerJ Computer Science, doi:10.7717/peerj-cs.1609_

## Round 0.1 · original submission · Major Revisions

The biggest handicap of this study is that no comparisons were made with large TSP problems. Besides the performance of proposed algorithms looks weak than traditional algorithms.

The article needs major revision in line with the referee's recommendations.

Reviewer 1 ·

Basic reporting

-Introduction section needs more detail. I suggest you definition the problem in detail.
-What are the strengths and drawbacks of the algorithms used in the study? If there are drawbacks, how did you take precautions in your study? Why were these two algorithms preferred over other algorithms? The reader should be enlightened on these issues.
-What are the 'some ways' mentioned in Line 391? The readers should be informed.

Experimental design

-Enriching the study with fluent diagrams will increase the quality in terms of readability and intelligibility.

Validity of the findings

-When we look at the literature, it is seen that there are many studies published in recent years regarding the content of this study. Therefore, this study lacks the studies published in recent years. A detailed literature review and comprehensive comparisons with the studies to be added should be made.
-In the conclusion section, a comprehensive comparison of the developed algorithm with other studies should be mentioned. Also, why was the deficiency mentioned in line 423 not considered in this study and will be developed in future studies? It will be important to emphasize this situation so that readers can understand it.

Reviewer 2 ·

Basic reporting

This work has several notions incorrectly used. For example, the "ant colony algorithm" and "simulated annealing algorithm" do not exist in general. Ant Colony Optimization and Simulated Annealing are metaheuristics, that are used for solving many problems. Each problem is approached over the years by many ACO and SA-based solving methods. Another example is the multiple usages of "TSP problems", which is incorrect. TSP means Traveling Salesman Problem. I think that the authors wanted to use "TSP instances". Bionic methods are a class of (meta)heuristic methods. Marco Dorigo invented the ACO, he is not the one who discovered the behavior of the biological ants. Ants do not "always choose the best route", but "usually".
Row 48. Solving the ant colony algorithm is incorrect, as an algorithm can not be solved.
Multiple references "the literature (...): are unusual. Please cite in a usual form "in (...) the authors propose...
Row 92. TSP has unclear origins, that go back to the 19th century.

Experimental design

Eq. 1 needs correction. It has dis(vi, vi+1). The mathematical formulation is unclear. I suggest using one of the traditional formulations, for example, the Dantzig–Fulkerson–Johnson (DFJ) formulation.
Eq. 2 is unclearly described, as Jh(i) is a set of ants. This set is in fact the same as allowedk.
Eq. 4. uses d with upper indices i and j, which in the text is described as d(ij).
Row 131 refers to starting end ending city, which in the TSP are the same. Please make a distinction between TSP and the problem you approach here. I think that you refer here to the subproblems from the proposed method, Also, DAC is the distance between A and j, which is not correct.
Eq. 8 describes the probability of acceptance of the new solution for the case when the optimization problem is a maximization one, which is not the case for the TSP.
It is not clear what 3DSA version is described in Table 3.
Row 229 please describe the criterion function.
Ro 248 "compute the ants of the current ant" is incorrect.
Row 253. The "less" sign is unusual. Why not use the "<"? It appears later on, again.
Row 268. How 1<rho0<0? Later on, again.
Row 323. Return to Step 5, I think.
Row 349. It is not clear what population you refer to.
Table 5. Why do the parameters have so diverse values? How did you choose these values?
Table 7. Usually, the algorithms used for comparisons are coded by the authors, in order to use the same computing environment and to assess also the computational costs. I suggest doing this.
Please briefly describe the instance solved in the final part (nodes, if it is a real distribution network).

Validity of the findings

The proposed method is tested on very small TSP instances. I suggest using the largest instances from TSPLIB.
The code is provided.
The conclusions must refer to the TSP instances tested.

Reviewer 3 ·

Basic reporting

1. Many researchers have studied both ant colony optimization (ACO) algorithms and simulated annealing (SA) algorithms for the TSP problem. This paper didn’t sufficiently review the current state of ACOs and SAs for the TSP problem. And the proposed algorithm was not compared with typical ACO algorithm (such as: Hassan Ismkhan, Effective heuristics for ant colony optimization to handle large-scale Problems, Swarm and Evolutionary Computation, http://dx.doi.org/10.1016/j.swevo.2016.06.006) and SA algorithm (Such as: Shi-hua Zhan et al. List-Based Simulated Annealing Algorithm for Traveling Salesman Problem, Computational Intelligence and Neuroscience, vol. 2016, Article ID 1712630, 12 pages, 2016. https://doi.org/10.1155/2016/1712630). As a result, the creativity, originality and competitiveness of the proposed method are not clear.
2. The writing and readability of the manuscript should be improved. There exit many errors, for example:
a. Lines 112 to 114.
b. Lines 133 to 134.
c. Lines 222 to 224.
3. The authors should describe the methods more clearly. And there exit apparent errors, such as:
a. Line 281
b. Line 300

Experimental design

The experiment part is not sufficient. The proposed algorithm was verified only on small-scale TSP instances. Authors should also verify the proposed algorithm on both medium-scale and large-scales TSP instances. The contribution of the components should be verified.

Validity of the findings

Overall, the main issue is that the algorithm lacks competitiveness in terms of performance. In fact, both ACO algorithm (Effective heuristics for ant colony optimization to handle large-scale Problems) and SA algorithm (List-Based Simulated Annealing Algorithm for Traveling Salesman Problem) have better performance than the proposed algorithm. Secondly, the experiments use small-scale TSP instances only and lack experimental verification of the contribution of the components of the proposed algorithm.

---

## Round 0.2 · Minor Revisions

The article needs minor revision in line with the referee's recommendations.

Reviewer 1 ·

Basic reporting

The relevant paper was re-reviewed after the revision request and the required changes in the basic reporting section were arranged by the authors.

Experimental design

The relevant paper was re-reviewed after the revision request and the required changes in the experimental design section were arranged by the authors.

Validity of the findings

The relevant paper was re-reviewed after the revision request and the required changes in the validity of the findings section were arranged by the authors.

Reviewer 2 ·

Basic reporting

1. There are several language issues. For example, capital letter missing at row 127, capital letter not needed at rows 76, 85, sentence starting with And at row 382, etc.
2. Rows referencing formula 6 contain not needed explanations of Jh(i) and allowedk - they disappeared in formula.
3. Rows 225-226 refers to dij which is the length of the current edge, not path
4. Formula 10 has an e, which is not explained.
5. Row 240: j is the next city, not the current one.
6. Row 280. Eq 19 is not correctly described.
7. Row 363. Again, "TSP problem", which means "Traveling Salesman Problem problem".

Experimental design

1. Please describe shortly the computational environment used for getting the results in Tables 1, 101 (I think it has to be 2), 2, 3, 101 (please rearrange the numbers).
2. The reference to the optimal solution in rows 466, 469, 490, 578, 586 and 614 must be changed, as the heuristic algorithms usually do not find the optimum solution. In my opinion, "the best solution found" is a correct expression.
3. Row 298. I do not understand why the parameters for AEACO differs from the one for ACO and EACO.
4. Table 4 contains the column "Size" which is not explained.

Validity of the findings

No major issues.

Additional comments

No additional comments.

---

## Round 0.3 · accepted · Accept

All necessary revisions have been made by the authors.

Reviewer 2 ·

Basic reporting

In my opinion, the authors did a full and good work in responding to the raised issues.

Experimental design

In my opinion, the authors did a full and good work in responding to the raised issues.

Validity of the findings

In my opinion, the authors did a full and good work in responding to the raised issues.

Additional comments

-